# Evaluation of an Oral Care Program to Improve the Oral Health of Home-Dwelling Older People

**DOI:** 10.3390/ijerph19127251

**Published:** 2022-06-13

**Authors:** Lina F. Weening-Verbree, Annemarie A. Schuller, Sytse U. Zuidema, Johannes S. M. Hobbelen

**Affiliations:** 1Research Group Healthy Ageing, Allied Health Care and Nursing, Hanze University of Applied Sciences, Petrus Driessenstraat 3, 9714 CA Groningen, The Netherlands; j.s.m.hobbelen@pl.hanze.nl; 2Center for Dentistry and Oral Hygiene, University Medical Center Groningen, A. Deusinglaan 1, FB 21, 9713 AV Groningen, The Netherlands; a.a.schuller@umcg.nl; 3TNO the Netherlands Organisation for Applied Scientific Research, Schipholweg 77-89, 2316 ZL Leiden, The Netherlands; 4Department of General Practice and Elderly Care Medicine, University of Groningen, University Medical Center Groningen, FA21, P.O. Box 196, 9700 AD Groningen, The Netherlands; s.u.zuidema@umcg.nl

**Keywords:** implementation, Oral Care Program, home care nursing, collaboration, older people, community-dwelling, daily oral care, support, formal home care nursing, attitude and knowledge

## Abstract

The aim of this study was to evaluate the impact of the implementation of an Oral Care Program on home care nurses’ attitudes and knowledge about oral health (care) and the impact on older people’s oral health. A pre–post study, without a control group, was conducted. A preventive Oral Care Program (OCP) was designed, focusing on home care nurses and older people, in collaboration with dental hygienists. Implementation was measured with questionnaires at baseline and after 6 months for home care nurses; for older people, implementation was measured at baseline and after 3 months with the Oral Health Assessment Tool and a questionnaire about oral (self) care between January 2018 and September 2019. Although the study design has limitations, the oral health of older people improved significantly after 3 months and the OCP was most beneficial for people with full dentures. The OCP improved knowledge and attitude of home care nurses. The program fitted well with the daily work routines of home care nurses. Individual-centered care plans for older people, education of home care nurses and the expertise of the dental hygienists have added value in home care nursing. Future implementations should focus on older people with natural teeth.

## 1. Introduction

Older people with decreased functional ability tend to have more oral health problems and poorer oral hygiene [1]. In general, oral hygiene is poor, periodontal problems are common and dental caries are often detected in community-dwelling, frail, older people, according to the limited data that are available [1,2,3,4]. Oral health is important, as multiple associations between oral health and general health exist. Inflammation in the oral cavity may disturb HbA1C levels in diabetes patients; oral inflammation may contribute to the manifestation of rheumatoid arthritis, and malnutrition and aspiration pneumonia are associated with poor oral health [5,6,7,8,9,10]. Oral health and inattention to oral hygiene by older people are associated with frailty and even mortality [2,10,11,12,13]. Recent studies show that the number of teeth missing is associated with increased frailty; older people who had oral health problems or were edentulous were more frail than older people with remaining natural teeth [3,14,15].

The number of regular dental visits decreases with age [16,17,18,19,20,21,22]. Unfortunately, not all dental clinics are proactive in preventing patient drop-out, nor are alternative care delivery systems such as dental home care with mobile care units available for all home-dwelling older people [17,18,20,21,22].

As the proportion of frail older people in the community is growing and more older people with complex care need to stay at their own home (instead of living in a nursing home) [23], the demand for formal home care is increasing [24]. Formal home care nursing in the Netherlands includes support of daily activities such as bathing and eating, and nursing care (e.g., giving medication, changing catheters and tube feeding). Additionally, (support with) daily oral care and monitoring oral health is included in formal home care nursing and nurses can refer the older people to oral health care providers if necessary [25].

Although home care nursing guidelines are clear about daily oral care as a primary component of activities of daily living care (ADL care), it is poorly integrated in the formal home care nursing system [19]. Person-centered oral care plans were not common in home care nursing and an interprofessional collaboration between the nursing profession and dental professionals was not established. Literature shows that home care nurses did not consider oral care as an aspect of fundamental daily care [26,27,28]. In addition, home care nurses indicated that they had no confidence in their ability to provide and manage oral care [26]. Home care nurses were more active in supporting hygiene of removable dentures, but guidance of home-dwelling older people with brushing natural teeth was not common [28]. It is suggested that oral care training, education and structured guidelines would improve the provision of oral care by home care nurses (self-efficacy) [26,27].

Limited literature is available regarding programs that aim to improve the oral health and hygiene of community-dwelling older people [27,29,30]. In order to improve oral health of community-dwelling older people and to reach out to this group of frail older people, it is necessary to develop, implement and evaluate an oral care program in home care nursing. This oral care program (OCP) needs to provide person-centered oral care and it should enhance a close collaboration between nurses and oral care providers [27]. Client-centered oral care is needed to enhance tools for oral care and oral health knowledge for older people. Therefore, the ‘Oral Care Program’ (OCP), originally entitled ‘Sûne Mûle’, was designed and implemented in a formal home care nursing setting. Our research aim was twofold—firstly, to evaluate the implementation of the OCP on home care nurses’ attitude and knowledge about oral health (care) and secondly, to investigate the impact of the OCP on older people’s oral health.

## 2. Materials and Methods

### 2.1. Design and Implementation of the ‘Oral Care Program’ (OCP)

The OCP focused on the three involved actors: community-dwelling older people, home care nurses and dental hygienists, and their mutual relationships, as illustrated in Figure 1. The OCP was designed in co-creation sessions with dental hygienists, dentists, general physicians, home care nurses and community-dwelling older people and was based on current guidelines in home care nursing [25]. The OCP included three main components: (1) raising awareness about oral health in older people and establishing a personal oral care plan; (2) education and practical instructions about oral care for the home care nurses; and (3) involvement of dental hygienists in the home care nursing teams.

The three stakeholders received several components:Home care nurses: (a) A general but tailored educational course about oral health (care) and the relationship with general health, based on the results of the baseline questionnaire about knowledge and attitudes of the home care nurses. The first course outlined the importance of oral health and the relationship with general health and wellbeing, and provided specific information about oral health problems in older people. (b) An in-depth module about the current guidelines for daily oral care including optional training of the practical skills. Apart from the guidelines [25] and practical information about (daily) oral care, the course included specific treatment and prevention of oral health problems in older people, e.g., dry mouth. The courses were given by a dental hygienist who is trained as a lecturer and dental hygienist at the school of Dental Hygiene Groningen. (c) Contact with a dental hygienist designated to the home care nursing team. The full courses and other detailed information about the OCP are available upon request by the first author.Home-dwelling older people in a home care setting: (a) A magazine with information about oral hygiene procedures, creating awareness of the importance of oral health (care). (b) A weekly calendar to support and give reminders of oral care with facts, reminders and suggestions. (c) Tailored oral care advice and instructions of a dental hygienist based on oral assessments and current guidelines in home care nursing. (d) Cards with tailored instructions for adequate oral health self-care [31]; for example, for older people with full dentures, the cards provided stepwise information about the daily oral care of the dentures and how to clean and store dentures. The cards could be put up in the bathroom. (e) Oral health care products. The products that were given were applicable to the older person. We provided toothbrushes, denture cases, denture brushes, interdental brushes, toothpicks and hourglasses to support brush time; according to the dental status, the appropriate products were provided.Dental hygienists: (a) Dental hygienists were allocated to a home care nursing team to answer questions or help with issues about oral health care.

### 2.2. Study Design

A pre–post study, with elements of an implementation study. The Revised Standards for Quality Improvement Reporting Excellence (SQUIRE 2.0) checklist was used to design and report this study (see Appendix A).

### 2.3. Sample and Recruitment

A convenience sample of six home care organizations in the northern part of the Netherlands (Friesland) were asked to participate, all agreeing to support their district nursing teams in joining the OCP. In the Netherlands, nursing staff with different levels of training work in home care: more practically educated, similar to nurse assistants, and also registered nurses who have obtained a bachelor’s degree [32]. We have chosen to use the definition ‘home care nurses’ (HCNs) as the collective term for all nursing staff members from the district home care nursing teams. The inclusion criteria for the home care nurses were that they should be involved in providing or assisting with general daily care for the clients. In practice, all home care nursing team members were included if the team agreed to participate in the OCP.

Secondly, the participating home care nursing teams were asked to approach eligible participants. Inclusion criteria were: aged 70 years or older, were using formal home care and signed informed consent. Exclusion criteria were: clients who were legally not capable to give informed consent or who were too ill to participate in the study.

### 2.4. Data Collection and Measurements

The implementation of the OCP started in January 2018 with an educational session in two home care nursing teams. Subsequently, more teams started, and by September 2019, all data were collected in 21 home care teams. Data on oral health were collected at baseline and after 3 months. Data on the implementation were collected at baseline and after 6 months. A schematic overview of the OCP and measurements are illustrated in Figure 2.

#### Implementation Assessment Instruments

##### Instruments Assessing the Impact of the OCP on Home Care Nurses


*MIDI—measurement instrument for determinants of innovations*


To evaluate the level of implementation, we used the measurement instrument for determinants of interventions (MIDI) of Fleuren et al. [33]. In our case, the OCP was the intervention to be evaluated. The MIDI was tailored for the OCP in accordance with the MIDI guidelines (Appendix A). The determinants of the MIDI are divided in four domains: characteristics of the user, of the intervention, the organization and the socio-political context.


*Questionnaire attitude and knowledge*


To assess the attitudes and knowledge of home care nurses, a questionnaire (Appendix A; in Dutch) was developed prior to the start of the OCP; it was based on previous questionnaires and current guidelines for oral care [25,34,35,36]. The questionnaire included questions about the (educational) background of the home care nurses, their opinion about performing or supporting clients with oral care and oral care materials. One question about attention for oral care asked for a rating of the attention on a 10-point scale—1 is no attention, 10 is a lot of attention. The knowledge questionnaire consisted of 29 statements with three answer categories—true/false/I do not know.

The MIDI and the attitude and knowledge questionnaire were distributed separately to all participating nursing team members using the program, Qualtrics [37]. The questionnaires were completed at baseline and 6 months after implementing the OCP.

##### Instruments Assessing the Impact on Oral Health of Older People


*Questionnaire older people—oral (self) care*


To assess oral (self) care and the performance of daily oral hygiene, a questionnaire (Appendix A) was compiled with items that derived from the national guidelines for daily oral hygiene [25]. The questionnaire consisted of questions about the background of the participant and procedures of cleaning teeth/dentures and about the last dental checkup they attended. This questionnaire was completed by the dental hygienist and the participant during the oral assessment with the OHAT.


*Oral Health Assessment Tool (OHAT)—older people’s oral health*


The clinical objective effects of the OCP on older peoples’ oral health and oral hygiene were operationalized using the OHAT [38]. The assessment covers the patient’s current oral health status and indicates the need for referral. The OHAT consists of eight items determining the condition of the lips, tongue, gums and tissues, saliva, natural teeth (if present), dentures (if present), oral cleanliness and dental pain, scored on an ordinal scale (0 = healthy, 1 = changes, and 2 = unhealthy). The sum score on OHAT is the sum of all the individual items, with a maximum of 16 in total. A sum score larger than 0 directs a referral to a dental professional. A higher sum score means that oral health and/or hygiene is poorer.

Seven registered dental hygienists with experience of working in the field of gerodontology were assigned to administer the OHAT in each participant at baseline and 3 months after the start of the OCP. The oral assessments were performed in the clients’ homes whilst they were sitting in a chair or bed. Ambient light was used to assess the oral cavity. The dental hygienists were unaware of the results of the baseline measurement during the follow-up examination and submitted the results one week after the completion of the first oral assessment.

### 2.5. Ethics

All data were processed anonymously, and privacy was respected according to the requirements of the Personal Data Protection Act. Approval was given by the Medical Ethical Committee of the University Medical Center Groningen for this study (study number 201700693). The study was judged not to fall under the provisions of the Medical Research Involving Human Subjects Act.

### 2.6. Analysis

#### 2.6.1. Data Analysis Regarding the Impact of the OCP on Home Care Nurses

Descriptive statistics (frequency distributions and means (standard deviations)) were used to describe the results of the MIDI and the attitude and knowledge questionnaire. The answers of the knowledge statements were dichotomized into correct and incorrect/do not know. Each correctly answered knowledge item scored 1 point, and totaled scores = sum score. A higher sum score indicated greater knowledge. The differences between the follow-up measurement and the baseline measurement were tested with Wilcoxon signed-rank tests or McNemar’s tests for data of nominal/ordinal origin and paired *t*-tests for continuous variables.

#### 2.6.2. Data Analysis Regarding the Impact of the OCP on Home-Dwelling Older People

The statistical analysis of the daily oral self-care questionnaires and the OHAT were performed by using frequency distributions for individual items and means for the sum scores (standard deviations). Mann–Whitney U tests/Wilcoxon Rank tests were performed to analyze the items of the OHAT scores at baseline and to compare the item scores after 3 months of the OCP. OHAT sum scores at baseline and follow up were calculated and paired sample *t*-tests were performed to compare means. All analyses were performed using Statistical Package for the Social Sciences for Windows Version 26.0 (IBM Corp., New York, NY, USA).

Furthermore, to analyze the OHAT data more deeply, we used mixed modeling [39] using the statistical language R (version 4.1.0) with a random effect per subject and fixed effects for sex, age, dental status, time, and team, the latter taken as factor. The F-test on their overall effects as well as the estimates of the effects were reported. The smaller teams from the same cities and team managers were clustered in the analysis, and thus, the number of teams in the analysis was 15.

The participants in the four groups of ‘dental status’ were divided into: older people with full dentures, natural teeth, a combination of natural teeth and dentures, and full dentures on dental implants. Paired sample *t*-tests were executed to compare the means of the different groups of ‘dental status’ in the OHAT sum scores to further explore the results of the OHAT scores. The level of significance was α = 0.05.

##### Validity and Reliability

To evaluate the level of implementation, we used the measurement instrument for determinants of interventions (MIDI) of Fleuren et al. [33]. To assess the attitude and knowledge of home care nurses, a questionnaire (Appendix A) was developed prior to the start of the OCP based on previous questionnaires and current guidelines for oral care [25,34,35,36]. To assess oral (self) care and the performance of daily oral hygiene, a questionnaire (File S2) was compiled with items that derived from the national guidelines for daily oral hygiene [25]. The OHAT is a reliable and valid instrument and designed to assess the oral health status of older people [38] and was translated to Dutch prior to the data collection; forward and backward translation and content validity was affirmed by the dental hygienists involved in OCP.

Furthermore, to avoid irregularities in the data collection and improve the reliability of the study, a strict research protocol was communicated with the home care nursing teams and the dental hygienists who performed the measurements. The dental hygienists were instructed on how to complete the clinical measurements prior to the baseline measurements.

## 3. Results

### 3.1. Response and Results of the MIDI Questionnaire

In total, 296 home care nurses (HCNs) divided over 21 home care nursing teams were eligible to participate in this study. Of these 296 HCNs, 100 (34%) completed the questionnaires. After six months, 286 questionnaires were sent and 99 (35%) were completed. Forty home care nurses completed both the baseline questionnaire as well as the follow-up questionnaire. In Appendix A, a table with an overview of the percentage scores on all items of the MIDI questionnaire is included.

The main findings of the MIDI concern:-Characteristics of the user: The HCNs and older people showed that HCNs expected that the OCP would make older people aware of the importance of oral health and most of the HCNs stated that they found oral health important for their clients. Additionally, HCNs considered it their job to observe the oral health of older people. After six months, even more HCNs than at baseline stated that the OCP was not time consuming for them, although the number of colleagues actually participating in the OCP decreased after six months.-The innovation (OCP): A positive attitude of HCNs towards the OCP at baseline and after six months was shown; more HCNs agreed that the OCP was not too complicated for them and the OCP fitted well with their daily work and was suitable for their clients.-The organization: HCNs care about the opinion of the manager, the organization, their colleagues/team members. After six months of working with the OCP, there were more agreements about participating in the OCP; HCNs were enabled to work with the OCP more, they knew more about the content of the OCP, and information was easily accessible. Additionally, HCNs were more positive after six months than at baseline about their ability to perform oral care and to assess older people’s oral (health) care.-Social-political context: The OCP matched well within current laws, rules and guidelines/procedures, and this scored more positively after six months.

### 3.2. Response on the Attitude and Knowledge Questionnaire for Home Care Nurses

The response of attitude and knowledge questionnaire at baseline was 58% (*n* = 173 HCNs of a total of 296 HCNs). The majority of the HCNs finished a 3-year nursing course. Most of the participants were female (*n* = 169).

The attitude and knowledge questionnaire after 6 months was completed by 112 HCNs (39% of 286 questionnaires). Both questionnaires were completed by 71 HCNs. The mean age of the HCNs was 43 years old (range 19–64 years). The mean working experience in home care nursing of the participants was 9.4 years (range 1–40 years).

No significant differences at baseline were determined in age, years of working experience, educational background and additional education about oral care between the HCNs who completed the knowledge and attitude questionnaires at baseline (*n* = 103) and the HCNs who completed both questionnaires (*n* = 71).

Most of the HCNs (80%, *n* = 139) received education about oral care in their initial training. Most of the HCNs did not follow additional courses or training about oral care (72%) and 64% of the participants indicated the need for more dental education at the start of the OCP. After 6 months, the demand for dental education was reduced to 34% of the HCNs, which was a statistically significant reduction (Z = −4.841, *p* < 0.001).

The HCNs attitudes changed slightly after 6 months, as shown in Table 1, but none of the attitude statement percentages changed significantly compared with the baseline percentages. Additionally, HCNs stated the two major challenges in oral care: 1. clients do not ask for support with oral care and 2. how important oral care is to clients.

After six months of the OCP, HCNs did not offer their support with oral care more often and clients did not ask for support with oral care more often than at baseline. HCNs did wear gloves more often than at baseline, but the results were not significantly different. HCNs stated that they themselves are responsible for oral health (care) changes in clients, but they indicated that oral health is also a shared responsibility with clients, their family and relatives, the general doctor and dental professionals; the percentages do not differ over time. The main reason that older people do not visit a dental professional (anymore) is that they have no complaints or they have full dentures, according to HCNs.

The HCNs reported that attention to oral care increased due to the OCP in six months. They scored 6.3 at baseline out of a maximum score of 10, and 6.8 after 6 months, which is a statistically significant difference (*t* (70) t = −2.652; *p* = 0.010).

Regarding the knowledge section of the questionnaire, the HCNs (*n* = 172) scored 22.7 questions (SD 3.1) of potentially 29 correct at baseline before the educational sessions were held. No one scored 29. After 6 months, the HCNs (*n* = 113) scored 24.3 (SD 2.3) and two HCNs scored 29. From 70 HCNs, both sum scores could be calculated and the results of the scores after six months improved significantly (*t* (71) = −5.70 *p* < 0.005).

Examples of knowledge items that were answered correctly more often after six months were about denture care (not to brush with toothpaste and not wear these during the night), the need for older people with dentures to go for check-ups, the effects of polypharmacy on oral health and the relationship between general health and oral health.

### 3.3. Response and Characteristics of the Older People

The eligible group consisted of approximately 1200 older people, which was the total number of people 70 years and over who were in care by the participating nursing teams at the time of this investigation. From all the older people approached by their HCNs, 190 clients agreed to participate in the study and provided informed consent. The mean age of the participants was 82.9 years (SD 7.7 years); 54 participants were men (28%) and 136 were women (72%). After informed consent was given, 18 participants could not be examined by the dental hygienist. The reasons for dropping out were: it was too difficult to make an appointment with the participants, some participants did not want to be visited by someone they did not know or they became too ill. These 18 participants were treated as dropouts in the analysis. The characteristics of the older people participating in this study are presented in Table 2.

The OCP was meaningful to 53% of the participants and participating in the OCP was ‘pleasant’ according to 43% of the participants. A minority of older people who participated in the OCP (17%) stated that the OCP did not offer them anything or that the OCP was too burdensome for them (2%).

### 3.4. Evaluation of Impact of the OCP on Patients’ Oral Health

The OHAT scores on the different categories were reported at baseline (t0) and after 3 months (t1) in Figure 3. The OCP impacted on the following items significantly: lips, tongue, mucosa, saliva and oral hygiene. Univariant analysis shows that denture brushes and soap were used significantly more frequently after three months of the OCP (*p* = 0.01), but the OCP did not significantly affect the oral hygiene behavior of participants with natural teeth.

Frequencies of the sum scores on OHAT at baseline and after 3 months were calculated. Participants scored lower sum scores after participating in the OCP: the mean OHAT sum score at baseline was 3.10 (SD 1.91) and 2.25 after 3 months (SD 1.63), a decrease of 0.86, which is a statistically significant improvement in oral health and hygiene (*t* = 5.631 (146), *p* < 0.001). More in-depth multi-level analyses with the variance F-test, given in Table 3 and Table 4, show significant effects for age, team, dental status and time, but not for ‘gender’ and ‘dental visit’. On average, the effect of ‘time’, after correction for other factors, resulted in an OHAT decrease of 0.82.

Additionally, the OHAT sum scores were calculated (post hoc) for the different dental status groups. At baseline, no significant differences were found between the groups. At t1, the analysis in Table 5 shows statistically significant differences in the group of older people with full dentures and in the group of older people with dentures and natural teeth in the three months participating in the OCP.

## 4. Discussion

### 4.1. The in Co-Creation Developed, Person-Centered and Tailor-Made OCP Resulted in a Significant Improvement in the Oral Health and Oral Hygiene of Older People after Three Months

Furthermore, the OCP resulted in a significant increase in knowledge in HCNs; it was confirmed that the OCP meets the need for education, as after six months, the need for education was significantly reduced. More importantly, the HCNs indicated that the OCP fits within their daily work schedule and was not considered to be too time consuming. On all four levels—the characteristics of the user, the innovation, the organization and the socio-political context—the OCP was aligned with their daily work and matched well with current guidelines and procedures. Remarkedly, the HCNs indicated that older clients are not accustomed to ask for assistance in oral care, and after implementing OCP, this behavior has not changed significantly. Denture hygiene improved—more older people cleaned their dentures with soap and denture brushes. Although overall oral health and oral hygiene improved significantly in older people due to the OCP, the group of older people with natural teeth benefited less from the OCP; it did not have a significant impact on the frequency of dental visits, but this is no surprise because the OCP focused on daily oral care and the follow-up time was only 3 months. However, visiting dental professionals for dental checkups should be encouraged in the future, as only half of the participants do this.

Although the OHAT sum scores improved significantly after three months, half of the older participants still scored a 1 or 2 score on oral hygiene, meaning that oral hygiene is not up to standard and needs improvement [25,34]. The results of our study are in accordance with other studies, showing poorer oral health in people with natural teeth in comparison with people with dentures [1,4,12]. The results of six months of the OCP with an average OHAT sum score of 2.25 are promising, considering the results of a recent study of dental intervention that reduced OHAT scores of older, community-dwelling participants to an average score of 2.83 [40]. Furthermore, the follow-up measurements were taken after 3 months of using the OCP; therefore, we have possibly only measured the short-term effects of the OCP.

Knowledge of HCNs improved significantly after six months of the OCP; however, there is still room for improvement of basic oral health care knowledge. Oral care education as an intervention for nursing staff in older people’s care showed positive results on attitude and knowledge in other studies [41,42,43]. We urge management and policymakers to consider this.

The OCP was implemented in home care nursing and in close collaboration with dental hygienists. Its impact on older people and on HCNs is shown in this study. A multi-modal program targeting several levels of care, such as the OCP, has higher value in contrast with programs targeting only one level, for example, only older people [40].

### 4.2. Limitations and Strengths of the Study

This study was designed as an implementation study and with no control group included; the effects on older people’s oral health and attitudes and knowledge of HCNs could therefore not be associated unequivocally with the OCP. On the other hand, the mixed-model analysis showed that the differences in oral health and hygiene were due to the effect of ‘time’, which can be assigned to the implementation of the OCP.

Only 71 HCNs completed the knowledge and attitude questionnaire at baseline and after six months, but to justify the results, no differences could be found in the group that completed both questionnaires and the group that only completed the baseline measurements. The MIDI questionnaires were indicated by HCNs as ‘difficult to complete’, resulting in only 40 questionnaires completed both at t0 and t1; therefore, the results were possibly not indicative of the opinion of the entire HCN sample.

The recruitment of older people from the home care nursing teams was via invitation by the HCNs, which may have resulted in a preselection of participants in advance and possibly not all eligible participants were asked to participate. Recruitment of older people who were home-dwelling was also a difficult matter in a similar study and the proportion of dropouts was a lot higher than in our study [40]. The older people are dependent on the HCNs for home care and may have participated in the OCP out of sympathy or because of their relationship with the HCNs, but the percentages of participants were similar between the teams (no clustering), and the clients performed oral care mainly themselves. The mean age of older participants in our study is high and a substantial number of participants have (partially) natural teeth, despite their age. Most of them do oral care themselves, which confirms the urgency for personalized care plans from the dental hygienists. However, one of the aims of the OCP was to encourage HCNs to support older people with oral care, and it seems the OCP did not succeed in that aspect. Furthermore, a convenience sample from one region in the Netherlands participated in this study, limiting the generalizability of the results. However, the Dutch home care nursing system works similarly nationwide and teams from multiple organizations were included, thus, the results of this study could be of value for other home care nursing teams. The results of the OCP cannot be transferred directly to other formal home care nursing teams; however, the results have added value for home care nursing teams who are ‘early adopters’ of oral care implementation projects, such as the OCP. Our study may be a ‘proof of concept’ because in the multilevel analysis (Table 3 and Table 4), the effect of time was significant. We suggest that in future, the OCP study should be repeated in order to support the current findings.

The OCP was implemented as a team intervention for home care nursing teams who were motivated to participate in the OCP, and this should be considered while interpretating the results. The OCP was implemented as one program, focusing on HCNs, older people and the involvement of dental hygienists, providing personal care plans, and education. The impact of the OCP cannot be assigned to a single aspect of the program and the results could be biased in a positive direction, since a collaboration with the dental hygienists was also part of the study design. Although we provided a team intervention, not all HCNs were part of the educational sessions, and this could have affected our study outcomes negatively. A possible explanation could be the high turnover rate of the HCNs [44], confirming the need for team interventions.

The dental hygienists who performed the screening of the OHAT on the participants were not blinded, therefore, the results could be biased in a positive direction. To avoid bias, all OHAT assessment forms were submitted within 1 week after the assessment. Although it cannot be ruled out that the assessor was able to remember the t0 OHAT scores 3 months later, this is highly unlikely.

The OHAT has no segment for dentures on implants; therefore, these data were reported in the item, ‘dentures’. In future research projects using the OHAT to assess dentures on implants, a separate segment should be made for implants to register the implant status. This information could not be included in the OHAT so far.

The data of this study were collected just before the COVID−19 pandemic. The COVID-19 pandemic reinforced the difficulties and barriers for older people to access dental care and researchers suggest that the COVID-19 pandemic will increase dental problems in frail older people [45,46]. This highlights the importance to implement oral care programs such as the OCP in home care nursing in the near future.

## 5. Conclusions

The OCP intervention improved the oral health of community-dwelling older people significantly after 3 months, especially for people with full dentures. The OCP fitted well into the HCNs daily work schedule, was not too time consuming, and improved both knowledge and attitudes of the HCNs. The person-centered care plans, education of HCNs and the expertise of the dental hygienists in the OCP have shown their added value in home care nursing; however, no control group was used to confirm the results of the OCP. Future implementations should focus more on supporting HCNs in the oral care of older people and adjustments may need to be made to the OCP to better accommodate older people with natural teeth.

## Figures and Tables

**Figure 1 ijerph-19-07251-f001:**
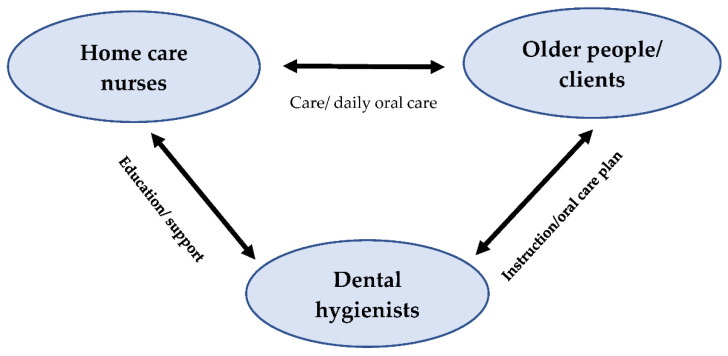
Role of dental hygienists, home care nurses and older people.

**Figure 2 ijerph-19-07251-f002:**
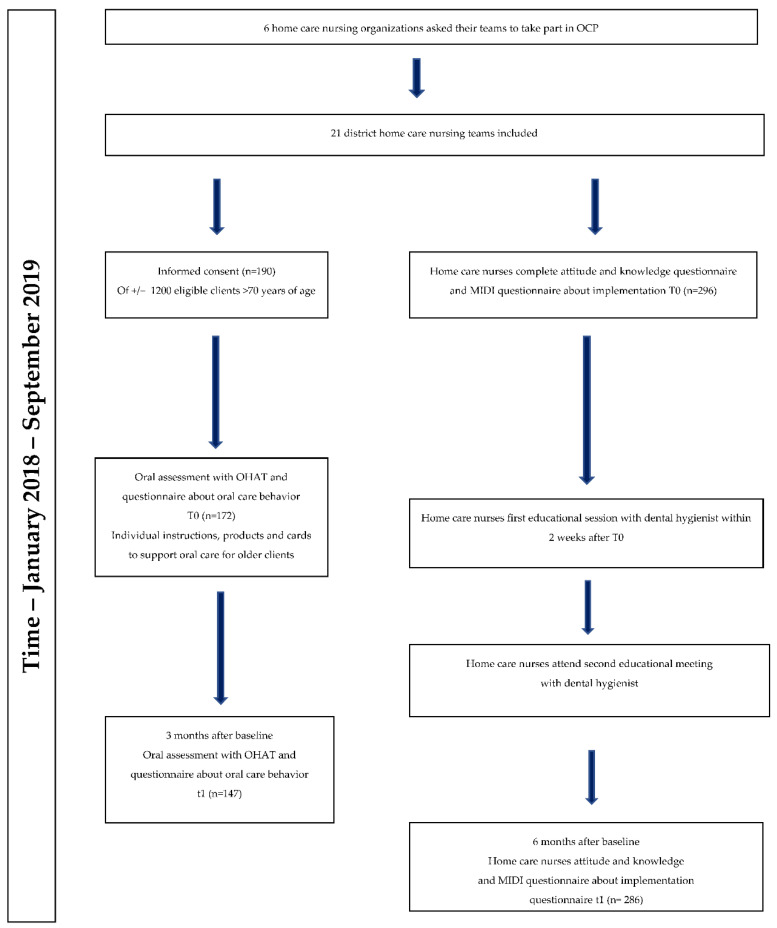
Schematic overview of the OCP and measurements.

**Figure 3 ijerph-19-07251-f003:**
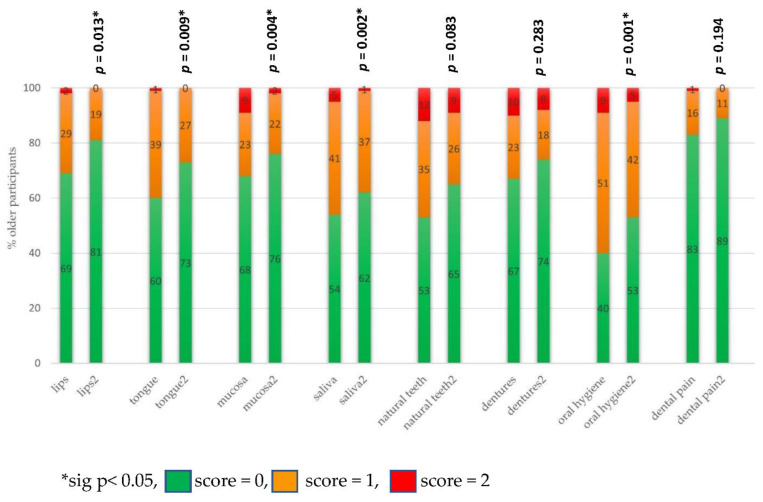
OHAT scores on different categories t0 and t1.

**Table 1 ijerph-19-07251-t001:** Completed attitude statements by HCNs at t0 and t1.

Statements—Attitude Items	T0 *n* = 173 (%)		T1 *n* = 114 (%)	
	Agree	Do Not Agree	Agree	Do Not Agree
I think clients think a healthy mouth is important	84	16	83	17
Our clients do not ask support for oral care	65	35	70	30
I think it is important to take care of clients’ oral care	100	0	97	3
I have enough skills to perform oral care	79	21	89	11
I have enough time to perform oral care	72	28	72	28
There are enough materials available to perform oral care (toothbrushes etcetera)	46	53	54	46
I think oral care should be part of the clients’ general personal care plan	92	8	92	8
I feel insecure to perform oral care for clients	8	92	7	93
I think it is an unpleasant task to perform oral care for clients	11	89	12	88
I think it is difficult to perform oral care for clients	16	84	15	85

**Table 2 ijerph-19-07251-t002:** Characteristics and oral status of older participants at baseline.

Characteristics	*n* = 172	*n* *	%
Mean age (SD)	82.9 (7.7)		
Gender, women	136		72
**Dental status**			
Natural teeth	27		16
Dentures	90		52
Dentures/natural teeth	35		20
Dental implants/dentures	20		12
**Dental visits**			
Visits a dental professional (yes)	76		44
Visited a dentist in the past 2 years (yes)	74		43
**Oral (self) care**			
Tooth brushing, two times a day or more	45	83	54
Tooth brushing, less than two times a day	38	83	46
Interdental cleaning daily (yes)	25	73	34
Denture cleaning with water and soap	26	145	18
Client does oral care him/herself	156		91

SD: Standard deviation, *n* *: the number of participants the item is applicable to.

**Table 3 ijerph-19-07251-t003:** Results of the variance F-test.

	Sum Sq	Mean Sq	F Value	*p* Value
Gender	0.263	0.263	0.162	0.688
Age	5.549	5.549	3.416	0.067
factor(dental status)	13.867	4.622	2.845	0.039 *
dental visits	6.194	6.194	3.813	0.052
factor(team)	43.389	3.099	1.908	0.029 *
factor(time)	14.026	14.026	8.634	0.000 **
factor(Team):factor(Time)	38.713	2.765	1.702	0.061

Sum Sq = sum of squares, Mean Sq = Mean Square, * sig *p* < 0.05, ** sig *p* < 0.01.

**Table 4 ijerph-19-07251-t004:** Multilevel model of OHAT scores.

	Estimate	Std. Error	df	*t* Value	*p* Value
(Intercept)	6.540	1.363	147.425	4.799	0.000 **
factor(Time)2	−0.821	0.152	154.928	−5.390	0.000 **
Gender	−0.135	0.257	150.563	−0.526	0.599
Age	−0.029	0.016	148.550	−1.787	0.076
factor(DentalstatusFac)1	−0.918	0.358	168.304	−2.561	0.011 *
factor(DentalstatusFac)2	−0.167	0.390	144.763	−0.429	0.669
factor(DentalstatusFac)4	−0.713	0.434	149.107	−1.643	0.102
DentVisit	−0.413	0.231	292.885	−1.790	0.075
factor(Team)1	−0.046	0.838	174.374	−0.055	0.956
factor(Team)2	0.012	0.550	143.240	0.022	0.983
factor(Team)3	−0.276	0.600	147.408	−0.460	0.647
factor(Team)4	−0.650	0.716	145.290	−0.908	0.366
factor(Team)5	0.148	0.627	144.668	0.237	0.813
factor(Team)6	−0.619	0.550	146.395	−1.126	0.262
factor(Team)7	−1.660	0.618	151.270	−2.686	0.008 **
factor(Team)9	0.119	0.558	144.741	0.213	0.831
factor(Team)10	0.696	0.598	149.620	1.165	0.246
factor(Team)11	0.490	0.704	141.697	0.697	0.487
factor(Team)12	−0.316	1.195	170.085	−0.265	0.792
factor(Team)13	−0.018	0.711	146.791	−0.025	0.980
factor(Team)14	−1.00	0.778	169.624	−1.289	0.199
factor(Team)15	0.635	0.691	145.441	0.919	0.359

Sum Sq = sum of squares, Std. Error = standard error, df = degrees of freedom, * sig *p* < 0.05, ** sig *p* < 0.01.

**Table 5 ijerph-19-07251-t005:** Results of the paired *t*-tests of the mean OHAT sum scores at baseline (T0) and after 3 months (T1) within oral status.

Dental Status	*n* at t0 and t1	Mean OHAT Sum Score T0	SD of Mean OHAT T0	Mean OHAT Sum Score T1	SD of Mean OHAT T1	MD	SD of MD	*t* Value	*p* Value
Natural dentition	27	3.30	2..20	2.85	2.14	0.44	1.91	1.21	0.24
Full dentures	75	2.85	1.89	1.85	1.40	1.00	1.82	4.750	0.00 **
Dentures and natural teeth	29	3.76	1.94	2.83	1.37	0.93	2.02	2.49	0.02 *
Dentures on implants	16	2.95	1.36	2.06	1.61	0.81	1.72	1.89	0.78

*n* = number of participants, SD = standard deviation, MD = mean difference, * sig *p* < 0.05, ** sig *p* < 0.001.

## Data Availability

The data presented in this study and detailed information about the Oral Care Program and educational sessions are available on request from the corresponding author. The data are not publicly available due to privacy or ethical reasons.

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
