# Peer review of "Evaluation of an Oral Care Program to Improve the Oral Health of Home-Dwelling Older People"

_ijerph, 2022, doi:10.3390/ijerph19127251_

Round 1

Reviewer 1 Report

I have reviewed the manuscript entitled: Evaluation of an Oral Care Program to improve the oral health of home dwelling older people.

In my opinion, I think the authors have done a good job. I consider this manuscript suitable for publication without changes.

Author Response

Response 1: thank you very much for reviewing our manuscript and we appreciate your response.

Reviewer 2 Report

This manuscript is focused on the impact of oral care programs on home care nurse’s knowledge and elderlies’ oral health. The topic is interesting but there were still some problems of contents, figures, tables, and concepts need to be deeply clarified and revised.

General Comment:

This manuscript is focus on the impaction of oral care program on home care nurse’s knowledge and elderlies’ oral health. The topic is interesting but there were still some problems of contents, figures, tables, and concepts need to be deeply clarified and revised.

Introduction:

1.    Please align the layout of the introduction, material and methods,   results, and discussion part.

2.    Line 64, mentioned that home care nurses indicated that they felt unable and uncomfortable providing oral care, the reference should be added.

3.    There was a lack of the hypothesis of the article.

Material and methods:

1.    Line 129, mentioned that nursing staff with different levels of training work in home care. The different levels must be explained in detail.

2.    The inclusion and exclusion part should be added more.

3.    The figure 2 layout needs to be cleaner and easier to understand. This figure should be reconstructing again.

4.    Line 184, mentioned that seven trained dental hygienists were assigned to administer the OHAT…, the definition of the “trained” should be clarified.

5.    The recruit age criteria are 70 years old, is there any reference to defined as the elderly age? Because from WHO definition of elderly is 65 years old.

6.    The lack of what content that dental hygienist educated the home care nurse, it’s a major problem, should be mentioned.

7.    The dependency of the participant must be mentioned because that might influence the efficacy and application of the oral care program.

8.    The oral health care products of the participant should be more useful and efficiency.

9.    The content of the cards should be informed.

Results:

1.    In table 2, please arrange neatly all the tables and figures to be consistent.

2.    In figure 3, the definition of “ * ” need to explained below the figure,  and some of the p-value didn’t lower than 0.05 still mark “ * ”, need to pay attention. The figure layout needs to be clearer and easier to understand.

3.    In table 3, the table layout needs to be clearer and easier to understand.

4.    In table 4, there was no dot of “p < 0.05”.

5.    Appendix B might add English version.

Discussion:

1.    Please explain the reliability and validity of the study, and the calculation method of the sample size should be clarified, as well.

Author Response

thank you for your review, we have added a response letter 

Reviewer 3 Report

This paper is presenting the results of a pre-post study without control group. The intervention consisted of a preventive oral care program, focusing on home care nurses and older people. The results showed that the oral health of older people improved significantly, and that the program improved the knowledge and attitude of home care nurses etc.

I found this study interesting as interventions to improve oral health of home dwelling older people is scarce. I agree that this is a very important field. Generally, I think the study is well performed and presented, using relevant questionnaires and sound methods.

However, I think the design of the study makes its findings questionable, mainly as no control group was used. As there are several moments in the implementation when the older people, the nurses, and the dental hygienists are interacting with the study designers there are high risk of exaggerating the effects. Even though the design and lack of control group is presented, I think this methodological weakness is not visible enough in the current version, especially not in the title and abstract.

The authors state that this is partly an implementation study. However, I still think it is quite far from showing real-world results.

So basically, I think the discussion section needs to present more of the weaknesses of the study. In the abstract I think it must be clearer that no control group exist, and the results and conclusions need to be related to this fact. You also state that the program was not “too time consuming”, but this should rather be analysed with data, preferable in a cost-effectiveness analysis.  

Author Response

(The authors gave the same response as above.)

Round 2

Reviewer 2 Report

Thank you, well done.

Reviewer 3 Report

The authors have revised the manuscript nicely and I think it is now ready for publication.